# Role of Bacteria-Derived Flavins in Plant Growth Promotion and Phytochemical Accumulation in Leafy Vegetables

**DOI:** 10.3390/ijms241713311

**Published:** 2023-08-28

**Authors:** Nivethika Ajeethan, Svetlana N. Yurgel, Lord Abbey

**Affiliations:** 1Department of Plant, Food, and Environmental Sciences, Faculty of Agriculture, Dalhousie University, Halifax, NS B2N 5E3, Canada; 2USDA, ARS, Grain Legume Genetics and Physiology Research Unit, Prosser, WA 99350, USA; svetlana.yurgel@usda.gov

**Keywords:** bacteria-derived flavins, plant growth promotion, riboflavin, *Sinorhizobium meliloti*, lettuce, kale

## Abstract

*Sinorhizobium meliloti* 1021 bacteria secretes a considerable amount of flavins (FLs) and can form a nitrogen-fixing symbiosis with legumes. This strain is also associated with non-legume plants. However, its role in plant growth promotion (PGP) of non-legumes is not well understood. The present study evaluated the growth and development of lettuce (*Lactuca sativa*) and kale (*Brassica oleracea* var. acephala) plants inoculated with *S. meliloti* 1021 (FL^+^) and its mutant 1021Δ*ribBA*, with a limited ability to secrete FLs (FL^−^). The results from this study indicated that inoculation with 1021 significantly (*p* < 0.05) increased the lengths and surface areas of the roots and hypocotyls of the seedlings compared to 1021Δ*ribBA*. The kale and lettuce seedlings recorded 19% and 14% increases in total root length, respectively, following inoculation with 1021 compared to 1021Δ*ribBA*. A greenhouse study showed that plant growth, photosynthetic rate, and yield were improved by 1021 inoculation. Moreover, chlorophylls *a* and *b*, and total carotenoids were more significantly (*p* < 0.05) increased in kale plants associated with 1021 than non-inoculated plants. In kale, total phenolics and flavonoids were significantly (*p* < 0.05) increased by 6% and 23%, respectively, and in lettuce, the increments were 102% and 57%, respectively, following 1021 inoculation. Overall, bacterial-derived FLs enhanced kale and lettuce plant growth, physiological indices, and yield. Future investigation will use proteomic approaches combined with plant physiological responses to better understand host-plant responses to bacteria-derived FLs.

## 1. Introduction

The challenges imposed by sustained climate change and global warming have had tremendous negative impacts on the global food supply chain and human health. Part of this challenge is caused by increased population growth and increased food and nutrition insecurity [1]. To feed this increased population, it is crucial that agricultural productivity must be remarkably increased within the next few years. Farmers worldwide are making efforts to increase food production either by increasing crop production acreages or by enhancing productivity on existing agricultural lands through inputs, such as fertilizer and irrigation, and adopting new methods, such as precision farming and artificial intelligence [2]. However, excessive usage of synthetic chemical fertilizers is harmful to the environment as it pollutes the soil, water, and air [3,4]. The main environmental impacts associated with fertilizers are nitrate leaching into ground water, emission of greenhouse gases, toxic heavy metal pollution in soils, and surface runoff of nitrogen (N) and phosphorus (P), leading to aquatic eutrophication [5,6,7,8].

Providing sufficient food for an increasing human population will require different strategies and approaches. The usage of plant growth-promoting (PGP) microbes as biofertilizers or biocontrol agents is a promising alternative to support eco-friendly and sustainable agricultural production systems. PGP microbes can enhance plant growth and protect plants from biotic and abiotic stresses through the facilitation of nutrient uptake from the soil and atmosphere, synthesis of specific compounds, and assisting the plant to tolerate and adapt to changes in environmental factors [9]. Several important bacterial characteristics, such as biological nitrogen fixation [10,11], phosphate solubilization [11,12,13], ACC (1-aminocyclopropane-1-carboxylic acid) deaminase enzyme activity [14,15,16], production of siderophores [17,18], and maintaining phytohormones level [17,19] are some traits that help PGP. Some recent research identified that bacterial-derived flavins (FLs) also have the PGP effect [20,21]. FLs are identified as one of the most chemically diverse prosthetic groups in biochemistry [22]. Riboflavin (RF, 7,8-dimethyl-10-ribitylisoalloxazine), commonly known as vitamin B2 is vital to all organisms and plays a vital role in oxidative metabolism. It is acknowledged that bacteria have functional pathways for the biosynthesis of RF [23,24].

Rhizobia are well-known PGP bacteria, that can infect or colonize the roots of legumes to form root nodules and are best known as symbiotic nitrogen-fixing bacteria. However, many reports suggested that rhizobia could also influence and colonize roots of non-legumes, as they do with legumes [25,26]. Previous studies reported that the PGP in non-leguminous crops, such as rice (*Oryza sativa*) [27,28], maize (*Zea mays*) [29], barley (*Hordeum vulgare*) and wheat (*Triticum aestivum*) [30,31], lettuce (*Lactuca sativa*) [32], pepper (*Capsicum annuum*) [33], tomato (*Solanum lycopersicum*) [33], and sunflower (*Helianthus annuus*) [34], forms association with several species of rhizobia. Rhizobia can also help to alleviate the adverse effects of environmental stresses on plants. For example, a study in rice showed that by increasing grain yield in drought- and saline-stressed fields with rhizobial inoculation [27]. Biswas [35] and Yanni [36] pointed out that the inoculation of rhizobia improves the uptake of several limiting plant nutrients in rice plants.

Several studies indicated that *S. meliloti* 1021 also enhances plant growth and development not only in leguminous crops but also in non-leguminous crops [26,37]. For example, inoculation in rice and wheat plants enhanced growth-related responses, such as increased rates of seed germination, seedling emergence, and vigor, increased vegetative biomass production, including elongation of seedling radical and plumule, expansion of root architecture, higher levels of chlorophyll, faster net photosynthesis rate, more dry matter, stomatal conductance, and increased rice and wheat grain yields [30,36,38,39].

The genetic organization of RF biosynthesis genes varies among bacteria. In bacteria, the enzymes encoded by the *ribBA*, *ribD*, *ribH*, and *ribE*/*C* genes convert one molecule of guanosine triphosphate (GTP) and two molecules of ribulose5-phosphate into RF [40]. Furthermore, in many bacteria, the enzymatic activities of RibB and RibA are combined in a single protein, which is encoded by a single *ribBA* gene [40]. The *ribBA* genes code for a putative bifunctional enzyme with dihydroxybutanone phosphate synthase and guanosine triphosphate (GTP) cyclohydrolase II activities, which catalyzes the initial steps of the RF biosynthesis pathway. It is believed that RibBA is a vital protein for enzymes responsible for catalyzing RF biosynthesis [41]. However, in *Sinorhizobium meliloti* strain 1021, the deletion of *ribBA* did not result in RF auxotrophy. The strain missing *ribBA* retained considerable GTP cyclohydrolase II activity but secreted a smaller amount of FLs, compared to the wild type, which ultimately affected bacterial root colonization [24]. Based on this study, we used *S. meliloti* strains 1021 (FL^+^) and *S. meliloti* 1021Δ*ribBA* (FL^−^) to test our hypothesis by using germination pouch assays and greenhouse potted-plant trials. The hypothesis of this study is that bacteria-derived FLs will improve the growth, physiology, yield, and phytochemicals in kale (*Brassica oleracea* var. acephala) and lettuce (*Lactuca sativa*).

## 2. Results

### 2.1. Effect of Bacterial-Derived FLs in Kale and Lettuce Seedlings Growth Parameters

Root scanner images (Figure 1) showed improved seedling morphology in FL^+^ inoculated seedlings (FL^+^ seedlings) compared to FL^−^ inoculated seedlings (FL^−^ seedlings). The inoculation with FL^+^ significantly (*p* < 0.05) increased the total root length, total root surface area, and total hypocotyl surface area of kale seedlings compared to FL^−^ seedlings (Table 1). More specifically, inoculation with FL*^+^*resulted in 19%, 20%, 19%, and 19% increases in kale seedling total root length, total root surface area, total hypocotyl length, and total hypocotyl surface area, respectively, compared to FL^−^ seedlings.

*S. meliloti* 1021 inoculation in lettuce seedlings exhibited significant (*p* < 0.05) increases in root and hypocotyl growth parameters, such as total length and total surface area (Table 2). The total root length and surface area were significantly (*p* < 0.05) increased by 13.7% and 14.3% in FL^+^ lettuce seedlings, respectively, compared to FL^−^ lettuce seedlings. Total hypocotyl length and surface area were higher (i.e., 7% and 4%, respectively) in FL^+^ lettuce seedlings compared to FL^−^ seedlings (Table 2).

### 2.2. Plant Growth, Physiology, and Yield

Kale plants were not significantly (*p* > 0.05) different in the number of leaves, plant height, and stem diameter between FL^+^ inoculated plants (FL^+^ plants) and FL^−^ inoculated plants (FL^−^ plants). However, the plant height and stem diameter were higher in FL^+^ plants than in the FL^−^ plants and control plants (Appendix A; Figure 2). Stem diameter was increased by 10% in FL^+^ plants than in FL^−^ plants (Appendix A). The kale leaf elongation rate was also higher in FL^+^ plants compared to the FL^−^ and control plants (Appendix A). Our results showed a significant (*p* < 0.05) difference in the stem diameter of kale between the FL^+^ plants and no inoculation controls (Appendix A). Chlorophyll fluorescence indices, Fv/Fo, and Fv/Fm were enhanced by bacterial-derived FLs (Appendix A). The results showed Fv/Fo and Fv/Fm were slightly increased in FL^+^ plants compared to the FL^−^ plants, even though there was no significant difference (*p* > 0.05) between treatments (Appendix A). Moreover, the fresh weight of kale was increased by 7% in FL^+^ plants compared to FL^−^ plants (Appendix A).

Lettuce plants inoculated with FL^+^ were shown to have increased plant height, stem diameter, chlorophyll indices, and fresh weight compared to the FL^−^ plants, even though they were not significantly (*p* > 0.05) different between treatments (Appendix A; Figure 2). The lettuce leaf elongation rate was also higher in FL^+^ plants than in the FL^−^ and control plants (Appendix A). The number of leaves, plant height, and stem diameter were significantly (*p* < 0.05) increased in FL^+^ plants compared to the control plants (Appendix A). Moreover, the fresh weight of the lettuce plants was significantly (*p* < 0.05) increased in the FL^+^ plants compared to the control plants (Appendix A). The fresh weight of the plants was increased by 5% with bacterial FLs. Significant differences were not observed in chlorophyll fluorescence indices (i.e., the maximum quantum efficiency of PSII and potential photosynthetic capacity) between treatments. However, the chlorophyll fluorescence indices were slightly increased in the FL^+^ plants compared to the other treatments (Appendix A).

### 2.3. Leaf Tissue Phytochemical Analysis

In kale leaves, chlorophyll *a* and *b* and carotenoid content in the leaves were significantly (*p* < 0.05) increased in bacteria-inoculated plants compared to control plants, although they were not significantly (*p* > 0.05) different between FL^+^ and FL^−^ kale plants (Figure 3A–C). FL^+^ plants showed that chlorophylls *a* and *b*, and carotenoid contents were 35%, 35%, and 26% increased, respectively, compared to the control plants (Figure 3A–C). Similarly, chlorophylls *a* and *b* were higher in FL^+^ plants than FL^−^ kale plants (Figure 3A–C). The total flavonoid content in kale leaves was significantly (*p* < 0.05) higher in FL^+^ plants by 28% and 12% more than in the FL^−^ and control plants, respectively (Figure 3D). The leaves of the FL^+^ plants had significantly (*p* < 0.05) higher total phenolic content than the FL^−^ and control plants (Figure 3E).

In lettuce leaves, even though chlorophyll *a*, chlorophyll *b*, and carotenoid contents were not significantly (*p* > 0.05) different between FL^+^ and FL^−^ plants, chlorophyll *a* and chlorophyll *b* were higher in the FL^+^ plants compared to the FL^−^ plants (Figure 4A–C). Total flavonoids and total phenolics content were significantly (*p* < 0.05) higher in the FL^+^ plants compared to the FL^−^ and control plants (Figure 4D,E). We observed that in FL^+^ plants, total flavonoids and total phenolics were 57% and 102% higher, respectively, than those found in the FL^−^ plants (Figure 4D,E).

### 2.4. Leaf Micro- and Macronutrient Elemental Compositions

Plant leaf tissue elemental analysis showed that micro and macro elements found in lettuce leaf tissues were influenced by bacterial-derived FLs (Table 3). The kale plants inoculated with FL^+^ had the highest phosphorus, potassium, magnesium, boron, and iron contents compared to the FL^−^ and control plants (Table 3). Nitrogen was higher in the FL^−^ plants than in the FL^+^ and control plants (Table 3).

In the lettuce leaves, the N content was approximately the same in the FL^+^ and control plants. However, N was increased by 9% in the FL^+^ plants compared to the FL^−^ plants (Table 3). Lettuce leaf tissue P was increased in the FL^+^ plants compared to the control and FL^−^ plants (Table 3). Leaf P content in the FL^+^ plants was 14% higher than in the FL^−^ plants. K was slightly reduced (4%) in the FL^+^ plants compared to the FL^−^ lettuce plants. Among the other nutritional, calcium, sodium, manganese, and zinc were increased by 2%, 24%, 18%, and 3% in the FL+ plants, respectively, compared to the FL^−^ plants (Table 3).

## 3. Discussion

Recent research showed that FLs play a key role in plant growth and promotion [20,24]. This is the first study to investigate the effect of bacterial-derived FLs in leafy vegetable plant growth, physiological functions, yield, and chemical composition. RF is an essential constituent in the coenzymes of flavin mononucleotide (FMN) and flavin adenine dinucleotide (FAD). FMN and FAD play key roles in several flavoprotein-mediated redox reactions in different metabolic pathways [42]. They also act as essential cofactors for a multitude of mainstream metabolic enzymes that mediate hydride, oxygen, and electron transfers. RF is easily converted enzymatically or photochemically into lumichrome [21,24], which is considered a potent molecule that stimulates plant growth and development [43]. According to Dakora [20], lumichrome and RF can influence ecological cues for sensing environmental stress, plant growth factors, signals for stomatal functioning, and act as protectants or elicitors of plant defense. While the mechanisms behind these functions remain unknown, it is possible that lumichrome induced the biosynthesis of classical phytohormones that caused the observed developmental changes in the plants [20].

Many cellular processes in plants directly depend on FLs, as flavoenzymes catalyze a broad spectrum of different reactions [44,45,46]. The present study showed that FLs facilitated plant growth and promoted metabolic activities from the seedling to the harvesting stage. The *S. meliloti* 1021 (FL^+^ strain) seedlings showed the highest root and hypocotyl growth compared to the *S. meliloti* 1021Δ*ribBA* (FL^−^ strain). Moreover, FL factors can both directly and indirectly enhance plant growth. For example, flavoproteins influenced *Arabidopsis* plant growth and development of hormonal levels, especially the metabolism of auxin and cytokinin [47].

A significant number of flavoproteins also play key roles in the primary and secondary metabolism of carbon, nitrogen, or sulfur assimilation, amine metabolism as well as the plant’s response to abiotic stress [47,48,49,50]. Interestingly, the results in the present study also evidenced that inoculation of FLs secreting *S. meliloti* 1021 improved plant physiological processes, mainly photosynthesis, leading to the enhanced yield of the kale and lettuce plants. Chlorophyll fluorescence indices are broadly used to determine the photosynthesis performance of plants [51]. Therefore, this finding can be attributed to increased Fv/Fo and Fv/Fm following the FLs secreting *S. meliloti* 1021 inoculation, although they were not significantly different (*p* > 0.05) between treatments. Chlorophyll has a key role in photosynthesis and the overall productivity of crops [52,53,54] and its level in plant leaves reveals the potency of the photosynthetic process. From the present study, quantitative analysis of photosynthetic pigments clearly showed that chlorophylls *a* and *b* (i.e., primary photosynthetic pigments) were higher in FL^+^ plants than in the FL^−^ plants and control plants. Carotenoids are accessory photosynthetic pigments that cannot transfer sunlight energy directly to the photosynthetic pathway, however, must pass their absorbed energy to chlorophyll and function as a photoprotector under excessive light incidence [55,56]. They also supply substrates for the biosynthesis of key plant growth regulators, such as abscisic acid and strigolactones [57,58]. The increased carotenoid content in kale and lettuce leaves with FLs secreting bacterial inoculation might have contributed to increased plant growth, physiological activities, and yield. Other than these photosynthetic pigments, plants require a balance of micro- and macronutrients to effectively perform photosynthesis [59]. Nitrogen (N) is one of the most important elements that determines plant growth [60,61]. Furthermore, N is found in the composition of numerous compounds, including amino acids, which make up proteins, nucleic acids, and nitrates and nitrites [62]. Previous studies have shown that leaf N content positively affects photosynthesis [63] through its influence on photosynthetic enzymes, pigment content, and the size, number, and composition of chloroplasts [64]. The present results also indicated that in lettuce, the percentage N was more increased in FL^+^ plants than in FL^−^ plants. In addition to N, we found out that phosphorus (P) in lettuce was enhanced by FL^+^. In kale, potassium (K) was increased in FL^+^ plants compared to the other treatments. Both P and K are essential elements that determine plant growth and productivity and are involved in several physiological processes, such as the activation of numerous enzymes, and the regulation of the cation–anion balance [65]. These physiological processes are essential for cell division and the development of the growing tip of plants [66].

Phenolic compounds are inherent in plants and are known as potential agents for preventing and mitigating many oxidative stress-related diseases [67]. Our results indicated that total phenolics and flavonoids in kale and lettuce were increased in FL^+^ plants. We demonstrated that FL-secreting bacterial inoculation in plants can be associated with enhanced phytochemical content in both kale and lettuce plants, as previously reported for herbal tea by Fu [67]. In addition, FL molecules, such as RF, can act as antioxidant agents, which could facilitate more benefits to plants through the enhancement of their leaf nutritional quality [68]. These findings suggest that the FLs secreting bacterial application can be considered as a greener and eco-friendly plant growth-promoting technique to increase the productivity and nutritional qualities of leafy vegetables and promote their health benefits.

## 4. Materials and Methods

This study was conducted in the Compost and Biostimulant Laboratory, Department of Plant, Food and Environmental Sciences, Faculty of Agriculture, Dalhousie University, Canada.

### 4.1. Germination Pouch Trial

#### 4.1.1. Bacterial Strains and Media

*Sinorhizobium meliloti* strains 1021 (FL^+^) [69] and 1021Δ*ribBA* (FL^−^) [24] were used in this study. *S. meliloti* 1021 strains were grown at 28 °C in minimum mannitol ammonium (MMNH_4_) medium [70].

#### 4.1.2. Inoculum Preparation and Seed Treatment

Kale (*Brassica oleracea* var. acephala ‘Dwarf Green Curled’) and lettuce (*Lactuca sativa* ‘Simpson’) seeds were purchased from Halifax Seeds (Halifax, NS, Canada). *S. meliloti* strains (FL^+^, FL^−^) were grown for 5 days in MMNH_4_ medium plates. Cultures were scrubbed and suspended in sterile water to prepare bacterial solutions with an OD_600_ of 0.6. The lettuce and kale seeds were sterilized with 10% sodium hypochlorite for 10 min and washed thoroughly three times with sterile distilled water (ddH_2_O). The cleaned seeds were treated by soaking in solutions of FL^+^, FL^−^, and sterile ddH_2_O, which was the control treatment, for 90 min. The treated seeds were air-dried under a laminar flow for 30 min. Ten seeds per treatment were placed in a 16.5 × 18 cm CYG germination pouch (Mega International, Roseville, MN, USA) with six biological replicates, and saturated with 25 mL of the respective solution per seed treatment.

#### 4.1.3. Seedling Growth Components

The pouches were incubated in the dark for two days at 22 °C, before being transferred into a growth chamber under a day/night temperature regime of 25 °C/20 °C, 16/8 h day/night illumination, light intensity of 300 μmol m^−2^·s^−1^, and relative humidity of 70%. The experiment was arranged in a completely randomized design with six replications and repeated twice. Every day, 3 to 5 mL of sterile ddH_2_O was added to maintain regular moisture, for nine days.

Root and hypocotyl morphological parameters were analyzed for 5 seedlings per pouch on the 9th day using a root scanning apparatus STD4800 Scanner, Epson Perfection V850 Pro, equipped with WinRHIZO™2000 software (Regent Instruments, Quebec, QC, Canada). The WinRHIZO™2000 software recognized scanned digital root and hypocotyl images and analyzed morphological traits, such as total lengths and total surface areas of root and hypocotyl [71,72,73].

#### 4.1.4. Statistical Analysis

All statistical analyses were performed using Minitab software version 20 (Minitab Inc., State College, PA, USA). Data obtained with different treatments were subjected to a one-way analysis of variance (ANOVA). Post hoc means were compared using Fisher’s least significant difference (FSD) at α = 0.05.

### 4.2. Greenhouse Potted-Plant Trial

#### 4.2.1. Greenhouse Environment and Experimental Design

Pot experiments were carried out from March to June and September to December 2022 in the Department of Plant, Food, and Environmental Sciences greenhouse as two trials with seven biological replicates. Seeds selection and sterilization processes were the same as described in Section 4.1.2. The seedlings of each plant species were raised for 4 weeks in a growth chamber with a day/night temperature regime of 24 °C/22 °C, 16/8 h d^−1^ illumination at a light intensity of 300 μmol m^−2^·s^−1^, and relative humidity of 70%. The seedlings were transplanted into 15 cm diameter plastic pots with a 10.5 cm depth. The mean day/night temperature cycle in the greenhouse during the experiment from transplanting to final harvest was 28 °C/20 °C with 71% relative humidity. Supplementary lighting was set to a 16 h day length cycle using a 600 W HS2000 high-pressure sodium lamp with NAH600.579 ballast (P.L. Light Systems, Beamsville, ON, Canada). The growing medium was a general-purpose Pro-mix BX potting mix and transplanted seedlings were acclimatized for a week under greenhouse conditions before treatment application. Treatments were inoculation with FL^+^ strain, FL^−^ strain, and a control (without inoculation). Inoculum was prepared as described in Section 4.1.2 and applied 5 mL per pot at 2-week intervals.

#### 4.2.2. Plant Growth, Physiology, and Yield Parameters

The number of green leaves for each plant was counted at 4 weeks after transplanting (AT). Plant height was measured from the apical meristem to the collar of the stem. The fourth leaf of each plant was tagged to estimate the plant growth rate by measuring the leaf elongation rate. Chlorophyll fluorescence indices were determined from three leaves per plant at 30 days AT using a chlorophyll fluorometer (Optical Science, Hudson, NH, USA). Leaves were first dark adapted for 25 min before the initial fluorescence yield (Fo) was measured, after that the maximum chlorophyll fluorescence (Fm) emitted was measured during a saturating light pulse. Variable chlorophyll fluorescence (Fv) was calculated as Fv = Fm − Fo. Fluorometric parameters, including maximum quantum efficiency (Fv/Fm), and potential photosynthetic capacity (Fv/Fo), were determined at each saturating pulse. Fresh weights of leaf samples from each treatment were recorded at the final harvest, i.e., 35 days AT, using a portable tabletop balance (VWR-2002P2, VWR International, Mississauga, ON, Canada). Harvested leaves were composed of leaf blades attached to petiole that had ≤5% blemishes or damage on the leaf surfaces. Those with blemishes or damage > 5% of the total leaf surface area were culled and not considered edible or marketable and were not included in the yield data. The leaves were dried in a 52100-10 Cole-Parmer mechanical convection oven dryer (Cole-Parmer Instrumental Company, Vernon Hills, IL, USA) at 65 °C to constant weights for 72 h, to determine leaf tissue micro and macro elements at the Nova Scotia Department of Agriculture Laboratory, Truro, NS, using inductively coupled plasma mass spectrometry (PerkinElmer Perkin Elmer 2100DV, Waltham, MA, USA).

#### 4.2.3. Chlorophylls a and *b*, and Carotenoid Determination

Chlorophylls *a* and *b*, and carotenoid contents in kale and lettuce leaves were estimated as described by [74,75], with some modification. Ground leaf tissues (200 mg) were separately homogenized in 2 mL of 80% acetone. The mixture was centrifuged at 13,000× *g* for 15 min and the absorbance of the supernatant was measured at 646.8, 663.2, and 470 nm against 80% acetone as a blank using a Bio-Tek Synergy H1 Hybrid Multi-Mode Reader and the Gen5 software application version 3.08 (BioTek^®^ Instruments, Inc., Winooski, VT, USA). Leaf chlorophylls *a* and *b*, and the total carotenoid contents of leaves were calculated by the following formulas and were expressed as a μg/g fresh weight (FW).
Chla (μg/mL) = 12.25 × A_663.2_ − 2.79 × A_646.8_
Chlb (μg/mL) = 21.50 × A_646.8_ − 5.1 × A_663.2_
Car (μg/mL) = (1000 × A470 − 1.8 × chla–85.02 × chlb)/198

#### 4.2.4. Total Phenolics Determination

Total phenolic content (TPC) was estimated using the Folin–Ciocalteu method, in terms of gallic acid equivalent (GAE) in mg/g of the extract, as reported by [76], with some modifications. A total of 200 mg of kale and lettuce ground leaf tissues were separately added to 2 mL of ice-cold 95% methanol and incubated at room temperature for 48 h under dark conditions. The mixture was centrifuged at 13,000× *g* for 5 min before 100 μL of the supernatant was added to 200 μL of 10% (*v*/*v*) Folin–Ciocalteau reagent. The mixture was incubated in the dark at 25 °C for 2 h, after the addition of 800 μL of 700 mM sodium carbonate (Na_2_CO_3_). The absorbance of the supernatant was measured at 765 nm, using 95% (*v*/*v*) methanol as the blank. TPC was calculated using a gallic acid standard curve and expressed in terms of GAE (mg) per FW (g).

#### 4.2.5. Total Flavonoid Determination

Total flavonoid content (TFC) was calculated using the aluminum chloride (AlCl_3_) colorimetric method, as described by Chang [77] and Pourmorad [78], with some slight modifications. Ground kale and lettuce leaf samples (200 mg) were separately mixed with 2 mL of ice-cold 95% methanol and centrifuged at 13,000× *g* for 15 min. The reaction mixture containing 1.5 mL of 95% methanol, 0.1 mL of 10% AlCl_3_, 0.1 mL of 1 M potassium acetate, and 2.8 mL of distilled water was added to 500 μL of the supernatant. The solution was incubated at room temperature for 30 min and the absorbance was measured at 415 nm, using 95% (*v*/*v*) methanol as the blank without AlCl_3_. For TFC determination, quercetin was used to make the standard calibration curve and it was expressed in μg quercetin/g FW.

#### 4.2.6. Statistical Analysis

All statistical analyses were conducted using Minitab software version 20 (Minitab Inc., State College, PA, USA). Data obtained with different treatments were subjected to a one-way analysis of variance (ANOVA). Post hoc means were compared using Fisher’s least significant difference (FSD) for plant growth, physiology, and yield parameter analysis and Tukey test for phytochemical analysis at α = 0.05. GraphPad Prism 9 software was used for visualization purposes.

## 5. Conclusions

This is the first study to examine the role of bacterial-derived FLs on plant growth and the development of green leafy vegetables. The results revealed that PGP from bacterial-derived FLs can enhance kale and lettuce growth and the development from the seedling to the harvest stage. The present results indicated that inoculation with a FL^+^ wildtype strain significantly increased the growth of kale and lettuce plants compared to inoculation with a FL^−^ mutant strain, which secrete fewer FLs than *S. meliloti* 1021. The total phenolics and total flavonoid contents in the kale and lettuce leaves were significantly increased with bacterial FLs secretion. Two of the primary elements, nitrogen, and phosphorus, were increased by the bacterial-derived FLs in lettuce. Moreover, inoculation with FL^+^ apparently improved plant growth better than FL^−^. This study has the great potential to aid the development of a new methodology for improving the health and productivity of agriculturally important crops through inoculation of bacterial-derived FLs. However, in future investigations, we will use proteomic approaches combined with plant physiological responses to better understand host–plant responses to bacterial-derived FLs. Moreover, it will be important to investigate how bacterial-derived FLs impact on central carbon metabolism in kale and lettuce plants.

## Figures and Tables

**Figure 1 ijms-24-13311-f001:**
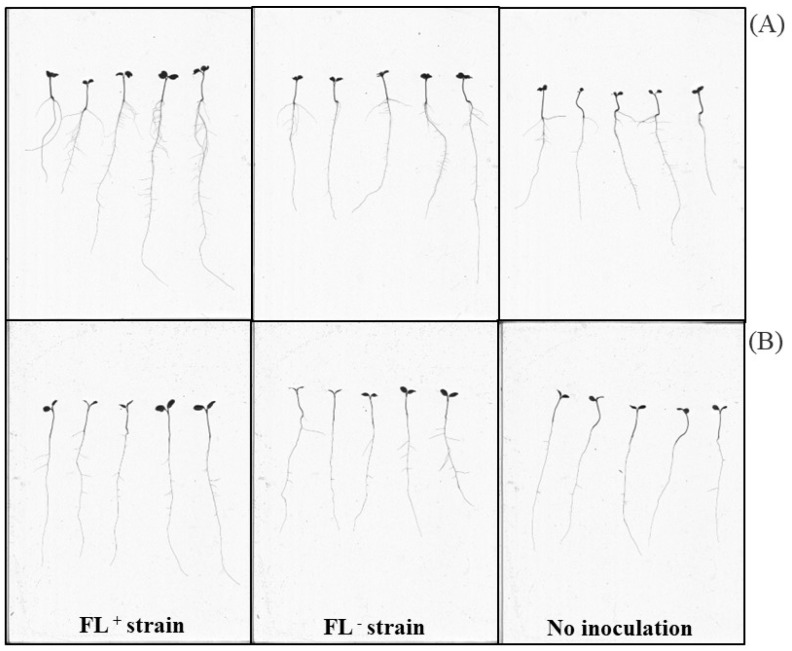
Effects of bacterial-derived FLs in kale (*Brassica oleracea* var. acephala) and lettuce (*Lactuca sativa*) seedlings. Photo comparison of the growth of (**A**) kale and (**B**) lettuce seedlings inoculated with *S. meliloti* 1021–FL^+^; *S. meliloti* 1021Δ*ribBA*–FL^−^; control: no inoculation.

**Figure 2 ijms-24-13311-f002:**
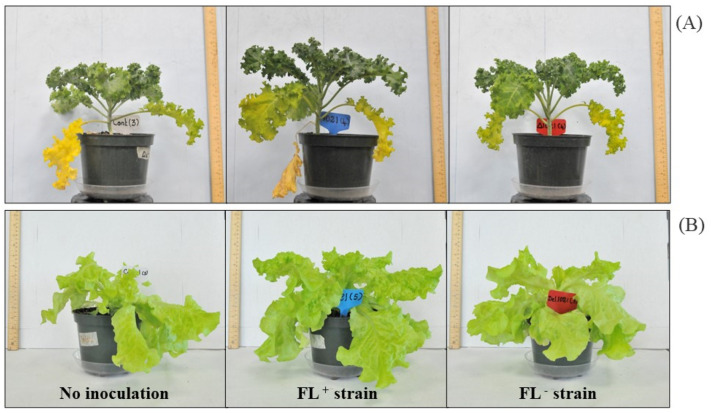
Effects of S. *meliloti* inoculation on the host plant. ((**A**) kale (*Brassica oleracea* var. acephala), (**B**) lettuce (*Lactuca sativa*)) growth. *S. meliloti* 1021: FL^+^ strain; *S. meliloti* 1021Δ*ribBA*: FL^−^ strain; control: no inoculation.

**Figure 3 ijms-24-13311-f003:**
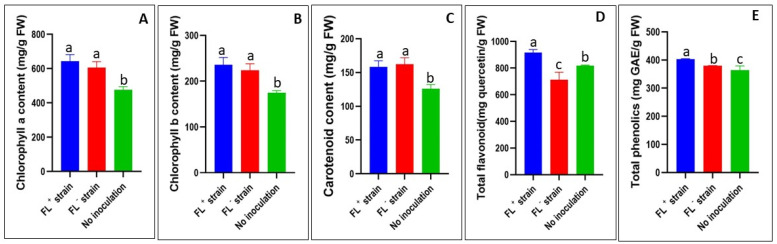
Effects of *S. meliloti* treatment on kale (*Brassica oleracea* var. acephala) leaf phytochemical properties ((**A**): chlorophylls *a*; (**B**): chlorophylls *b*; (**C**): carotenoid; (**D**): total flavonoid; (**E**): total phenolics). *S. meliloti* 1021: FL^+^ strain; *S. meliloti* 1021Δ*ribBA*: FL^−^ strain; control: no inoculation. Different alphabetical letters denote significant differences (*p* < 0.05) between treatment means by Tukey post hoc test.

**Figure 4 ijms-24-13311-f004:**
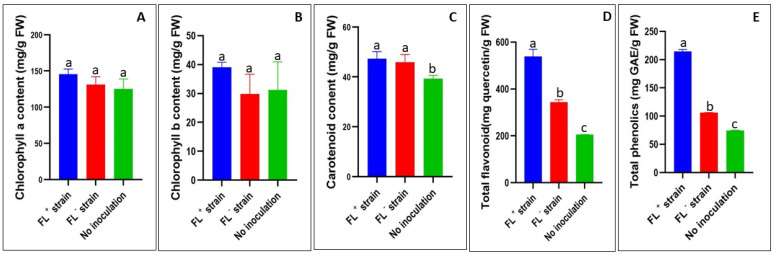
Effects of *S. meliloti* treatment on lettuce (*Lactuca sativa*) leaf phytochemical properties ((**A**): chlorophylls *a*; (**B**): chlorophylls *b*; (**C**): carotenoid; (**D**): total flavonoid; (**E**): total phenolics)*. S. meliloti* 1021: FL^+^ strain; *S. meliloti* 1021Δ*ribBA*: FL^−^ strain; control: no inoculation. Different alphabetical letters denote significant differences (*p* < 0.05) between treatment means by Tukey post hoc test.

**Table 1 ijms-24-13311-t001:** Effects of bacterial-derived FLs on kale (*Brassica oleracea* var. acephala) seedling growth parameters.

Treatment	Total Root Length (cm)	Total Root Surface Area (cm^2^)	Total Hypocotyl Length (cm)	Total Hypocotyl Surface Area (cm^2^)
FL^+^ strain	30.17 ± 2.56 a	11.81 ± 1.26 a	4.87 ± 0.86 a	2.86 ± 0.24 a
FL^−^ strain	25.35 ± 2.41 b	09.84 ± 1.52 b	4.39 ± 0.44 ab	2.58 ± 0.13 b
No inoculation	20.56 ± 2.32 c	10.38 ± 1.33 ab	3.68 ± 0.50 b	2.49 ± 0.14 b

*S. meliloti* 1021: FL^+^ strain; *S. meliloti* 1021Δ*ribBA*: FL^−^ strain; control: no inoculation. Different alphabetical letters denote significant differences (*p* < 0.05) between treatment means by Fisher’s least significant difference test.

**Table 2 ijms-24-13311-t002:** Effects of bacterial-derived FLs on lettuce (*Lactuca sativa*) seedling growth parameters.

Treatment	Total Root Length (cm)	Total Root Surface Area (cm^2^)	Total Shoot Length (cm)	Total Shoot Surface Area (cm^2^)
FL^+^ strain	20.13 ± 0.83 a	9.39 ± 0.38 a	5.98 ± 0.75 a	3.07 ± 0.22 a
FL^−^ strain	17.70 ± 0.79 b	8.22 ± 0.85 b	5.59 ± 0.61 a	2.95 ± 0.18 a
No inoculation	18.17 ± 0.50 ab	8.32 ± 0.95 b	5.15 ± 0.79 a	2.92 ± 0.25 a

*S. meliloti* 1021: FL^+^ strain; *S. meliloti* 1021Δ*ribBA*: FL^−^ strain; control: no inoculation. Different alphabetical letters denote significant differences (*p* < 0.05) between treatment means by Fisher’s least significant difference test.

**Table 3 ijms-24-13311-t003:** Effect of bacterial-derived FLs on leaf tissue micro- and macronutrients in kale (*Brassica oleracea* var. acephala) and lettuce (*Lactuca sativa*).

	Kale					Lettuce			
Treatment	FL^+^ Strain	FL^−^ Strain	No Inoculation	CV %	FL^+^ Strain	FL^−^ Strain	No Inoculation	CV %	Reporting Limit
Nitrogen (mg/mL)	19.70	20.80	19.20	4.113	14.00	12.80	14.10	5.306	00.20
Calcium (mg/mL)	16.79	16.37	16.85	1.569	10.79	10.62	10.26	2.563	00.02
Potassium (mg/mL)	21.82	21.64	20.37	3.715	22.29	23.34	23.64	3.070	00.15
Magnesium (mg/mL)	02.91	02.87	02.89	0.692	03.37	03.76	02.89	13.05	00.02
Phosphorus (mg/mL)	04.47	04.38	03.98	6.099	03.97	03.47	03.67	6.796	00.01
Sodium (mg/mL)	00.51	00.60	00.49	10.99	06.13	04.93	04.57	15.68	00.15
Boron (mg/L)	24.19	22.72	22.08	4.704	11.25	13.37	11.84	9.003	10.00
Copper (mg/L)	7.300	5.770	ND	16.56	ND	ND	ND	NA	5.000
Iron (mg/L)	50.53	50.14	43.11	8.713	41.20	45.00	37.43	9.185	5.000
Manganese (mg/L)	87.50	97.02	89.38	5.523	65.70	55.72	49.38	14.45	10.00
Zinc (mg/L)	51.87	47.45	53.03	5.798	33.81	32.94	34.87	2.853	2.000

*S. meliloti* 1021: FL^+^ strain; *S. meliloti* 1021ΔribBA: FL^−^ strain; control: no inoculation. CV %: percentage of coefficient of variation between treatments. ND: the value is below the reporting limit.

## Data Availability

The data for this study are available from the corresponding author upon reasonable request.

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
