# Peer review of "Role of Bacteria-Derived Flavins in Plant Growth Promotion and Phytochemical Accumulation in Leafy Vegetables"

_ijms, 2023, doi:10.3390/ijms241713311_

Round 1

Author Response

Thank you for your suggestions and comments, please see the attachment.

Reviewer 2 Report

Brief Summary

The manuscript ijms-2559931 is an interesting study on the study of the role of Sinorhizobium meliloti 1021-derived flavins in plant growth promotion and phytochemical accumulation in lettuce and kale.

The study is relevant and can add knowledge to the field. The experiments were conducted with valid methodologies. The quality of data handling and manuscript preparation is good. The manuscript needs minor improvements.

See specific comments below.

Specific comments

Title: The title does not reflect the content of the manuscript. I find it too generalist. You should provide a specific title focusing on the content of the manuscript.

Abstract: The summary of the main findings should be improved.

Introduction: The introduction correctly places the study in the context and considers a good number of references. However, some issue should be addressed.

o   Add the specific hypothesis being tested.

o   Add the brief methodological approach used.

o   References stop at 2022, more 2023 references should be provided.

Materials and Methods: The authors described the methods used clearly. Some minor issues:

·         Add results expressions in each section.

·         Provide standard curves equations when used.

·         The statistical analysis section should be moved to the end of the methods section. The description should also be improved. Were the data processed with the normality test?

·         The descriptions of “Plant growth”, “physiology”, and “yield” parameters should be divided into three different sections for clarity.

Results and discussion: The results description is clear and supported by appropriate figures.

Discussion: Authors correctly discussed the results considering a good number of previous studies. In this section the cited works also stop at 2022, more 2023 references should be provided.

Conclusions: The section is informative and supported by the findings obtained. Some specific future research directions should be provided in addition to lines 385-387.

Other comments:

·      Some minor typos are present in the text (e.g., punctuation, number and titles of sections formatting).

Author Response

Thank you for your suggestions and comments, please see the attachment
